# The Assessment of Fatigue in Rheumatoid Arthritis Patients and Its Impact on Their Quality of Life

**Waseem R. Dar** [1], **Irfan A. Mir** [1], **Summra Siddiq** [2], **Mir Nadeem** [3,*] and **Gurmeet Singh** [4]

1   Department of Medicine, School of Medical Science and Research, Greater Noida 201306, India; waseemramzan96@gmail.com (W.R.D.); drirfan765742@gmail.com (I.A.M.)
2   Medicine, New Cross Hospital, Wolverhampton WV1 1JA, UK; summra.siddiq1@nhs.net
3   Department of Medicine, King Khalid University, Abha 62529, Saudi Arabia
4   Department of Medicine, Government Medical College, Jammu 180001, India; mirirfan76@rediffmail.com
*   Correspondence: mirnadeem44@gmail.com

**Abstract:** Introduction: Rheumatoid arthritis (RA) is a common autoimmune illness that manifests mostly as chronic, symmetric, and progressive polyarthritis with a global frequency of 0.3–1.0%. RA is a disease that affects people all over the world. In India, the prevalence is estimated to be 0.7%, with around 10 million persons suffering from RA. Most people with rheumatoid arthritis experience fatigue on most days, with over 70% experiencing symptoms similar to chronic fatigue syndrome. Patients rate fatigue as a top priority and believe this unmanageable symptom is ignored by clinicians; a systematic review shows the biological agents for RA inflammation have only a small effect on fatigue. Fatigue predicts and reduces the quality of life, and it is as difficult to cope with as pain. Physicians have traditionally concentrated on the inflammatory aspects of the illness (e.g., synovitis), whereas RA patients have prioritized pain, exhaustion, sleep difficulties, and other quality-of-life issues. Aims and Objectives: The basic aim of the study was to access the incidence of fatigue in rheumatoid arthritis and evaluate its impact on the quality of life in these patients using the MAF scale (multidimensional assessment of fatigue) after prior permission for the first time in an Asian population. Results: A total of 140 subjects and 100 controls were included in the study. Age was closely matched between the study subjects and controls. Among study subjects with the disease, 94 (67%) had a disease duration $\leq$ 5 years, 26 (19%) had a disease duration between 6–10 years, 10 (7%) had a duration of 11–15 years and 10 (7%) had >10 years disease duration. Among the sample, 31 (25%) study subjects had a DAS score $\leq$ 4.0, 63 (50%) study subjects had a DAS score (disease activity score) between 4.01 and 6.0, and in the remaining 31 (25%) study subjects, the DAS score was >6.0. The mean DAS score among study subjects was 4.96, and the study subjects had a mean activity of daily living (ADL) score of 11.64; controls had a mean score of 2.42 with a statistically significant *p*-value. The global fatigue index was higher in study subjects, with a mean of 33.16 in contrast with a mean of 14.41 in the controls with a significant *p*-value. Conclusion: Our study fatigue was a persistent problem, despite treatment. The median level of fatigue experienced by study subjects with RA was high. Therefore, as persistent fatigue is associated with functional loss, fatigue in RA remains an 'unmet need' and continues to be ignored by clinicians.

**Keywords:** rheumatoid arthritis; fatigue; disease activity; polyarthritis; inflammatory

## 1. Introduction

Rheumatoid arthritis (RA) is a common autoimmune illness that manifests mostly as chronic, symmetric, and progressive polyarthritis, with a global frequency of 0.3–1.0% [1]. Women are up to five times more likely to get RA than males, and those over the age of 40 have the highest prevalence rates [2]. People from all around the world are afflicted by RA. In women over the age of 70, the overall prevalence of RA increases to 5%. Around 10 million people in India are thought to have RA, making the prevalence there 0.7% [3].

Over 70% of persons with rheumatoid arthritis report having symptoms like chronic fatigue syndrome on most days [4,5].

Fatigue is typically characterized as a state of exhaustion and weakness accompanied by a sense of weariness, drowsiness, and irritability, as well as a cognitive component [6]. Physiological exhaustion that occurs after strenuous physical exertion provides a signal to the body to rest so that the depleted tissues (that is, the muscles) can be rescued. Pathological fatigue, unlike normal exhaustion, does not improve with rest. Most acute and chronic inflammatory disorders, including arthritis, cause this type of fatigue. Fatigue is considered an extra-articular symptom, and it is the most prevalent complaint among RA patients, outnumbering pain by 42 to more than 80% [7]. Social variables, gender, cognition and emotion, sleep disturbances, pain, and environmental factors may all play a role in fatigue onset [8]. Because fatigue is a symptom of subjective experience, the most frequent descriptive method is self-assessment.

Fatigue is definitely a sign of rheumatic diseases. According to OMERACT [9], fatigue is an essential outcome to examine in rheumatoid arthritis (RA), and it has been linked to the Disease Activity Score 28 (DAS28), the ESR, and the Clinical Disease Activity Index [10]. Patients describe fatigue as a major source of anxiety, on par with pain, as overpowering, uncontrolled, and unaddressed by physicians. Significant fatigue affects up to 70% of RA patients, affects the quality of life, and is just as difficult to manage as pain. Because fatigue has been identified as an unmet need among RA patients, an international agreement has emerged that fatigue must now be assessed in all RA clinical studies. [11]. Fatigue has a wide range of effects on patients' lives and is a significant result for many, although it is presently not one of the seven universally agreed-upon key outcome measures in RA clinical studies [12,13]. Fatigue is cited by people with RA as the primary cause of job loss, affecting more than 60% of working patients and resulting in a production loss of more than GBP 650 million [14]. Patients rate fatigue as a top priority and believe that this unmanageable symptom is ignored by clinicians [12]; a systematic review shows that biological agents for RA inflammation have only a small effect on fatigue [15]. Fatigue predicts activities of daily living (ADLs) and reduces the quality of life, and it is as difficult to cope with as pain. Physicians have traditionally concentrated on the inflammatory aspects of the illness (e.g., synovitis), whereas RA patients have prioritized pain, fatigue, sleep difficulties, and other quality-of-life issues [16]. Fatigue in rheumatoid arthritis is now the most evolving topic in rheumatology, and our study is the first of its kind in the Asian subcontinent; it will be a stepping stone in the evaluation of the geographical variability of fatigue in rheumatoid arthritis.

## 2. Aims and Objectives

The basic aim of the study was to access the incidence of fatigue in rheumatoid arthritis and evaluate its impacts on the quality of life in these patients using the MAF scale (multidimensional assessment of fatigue), being used after prior permission for the first time in Asian population.

## 3. Material and Methods: [Study Design]

Our present study, entitled "Assessment of Fatigue in Rheumatoid Arthritis and its Impact on Quality of Life", is an observational, case–control one-point study that was conducted from November 2017 to October 2018. The protocol was reviewed and approved by the GMC Jammu ethical committee under order no. PGRP/2017/MDGMC. The study subjects were included after giving written informed consent and after receiving a full explanation of the nature and purpose of the study before participating in the study.

## 4. Participant Inclusion Criteria

Study subjects diagnosed with rheumatoid arthritis according to the 2010-ACR-EULAR classification criteria for rheumatoid arthritis aged 16 years and above were included in the study as cases and compared with age- and sex-matched controls attend-

ing medicine/ rheumatology OPD having health conditions other than musculoskeletal complains in GMC Jammu.

## 5. Exclusion Criteria

Patients who need hospital admission, critically ill patients, pregnant or lactating women, a diagnosis other than RA like osteoarthritis, ankylosing spondylitis, systemic sclerosis, chronic kidney disease, or diabetes mellitus.

## 6. Data Collection

The demographic profiles of all the study subjects were noted, and subjects were enrolled after giving written informed consent. A detailed history of RA and any other coexistence disease was obtained. A detailed physical examination of the study subjects was also performed.

Patients were on treatment for rheumatoid arthritis with DMARDS mainly, and disease activity was measured with the DAS 28, the ESR, and online calculators using the variables tender joint count, swollen joint count, and erythrocyte sedimentation rate. The degree of fatigue was assessed with the Multidimensional Assessment of Fatigue (MAF) [17,18] scale in both study subjects and controls. The MAF is a 16-item scale that measures fatigue according to four dimensions: degree and severity, distress that it causes, the timing of fatigue (over the past week, when it occurred, and any changes), and its impact on various activities of daily living (household chores, cooking, bathing, dressing, working, socializing, sexual activity, leisure and recreation, shopping, walking, and exercising). The global fatigue index (GFI) is a measure that quantifies five dimensions of fatigue from the Multidimensional Assessment of Fatigue instrument into one score and a newly scored item 15; Item 16 of MAF is not included in the GFI. A higher score indicates more severe fatigue, fatigue distress, or impact on activities of daily living.

The activity of daily living (ADL) score was used to look for the quantum of disability in RA patients. The ADL score consists of 12 items related to daily activities. It tests whether the patient can perform daily activities independently or whether they are unable to do so.

The quality of life of the subjects was assessed with the World Health Organization Quality of Life BREF (WHOQOL-BREF). The WHOQOL-BREF produces a quality-of-life profile with four domain scores (physical, psychological, social, and environmental). The four domain scores denote an individual's perception of quality of life in each particular domain. The domain scores are scaled in a positive direction, i.e, higher scores denote a higher quality of life. The mean score of items within each domain was used to calculate the domain score. The raw scores were then transformed to a 0–100 scale. The MAF scale was used to assess fatigue in RA patients. It includes questions about fatigue and the effects of fatigue on daily activities [17,18].

## 7. Statistical Analysis

All the data obtained from the subjects of the study group were noted in a Pro-forma especially designed for this purpose. The data were analyzed using appropriate statistical techniques in Open Stat software. After extraction, the data were revised, coded, and fed to statistical software IBM SPSS version 22 (SPSS, Inc., Chicago, IL, USA). All statistical analyses were conducted using two-tailed tests. A *p*-value less than 0.05 was statistically significant. Comparative analysis between study cohorts was conducted for all variables including patients' biodemographic data, quality of life, DAS28, ESR score, ADL, and MAF was used based on cross-tabulation. The significance of the relationships in cross-tabulation was tested using the Pearson chi-square test for categorical parameters, while scale variables were compared using the independent samples t-test.

## 8. Results

A total of 140 study subjects and 100 controls were taken into the study with ages closely matched with controls. The study subjects had a mean age of 47 years compared

with the control mean of 42 years with a *p*-value of 0.073. The distribution of patients according to age is shown in Table 1 below.

**Table 1.** The distributions of patients and controls according to age (*n* signifies the number of subjects and controls).

| Age (Years) | Patients *n* = 140 | Control *n* = 100 |
|---|---|---|
| ≤30 | 14 (10.00) | 23 (23.00) |
| 31–40 | 27 (19.29) | 24 (24.00) |
| 41–50 | 44 (31.43) | 26 (26.00) |
| 51–60 | 37 (26.43) | 20 (20.00) |
| >60 | 18 (12.86) | 7 (7.00) |
| Range | 17–78 | 19–79 |
| Mean age ± SD | 47.45 ± 12.08 | 42.44 ± 12.97 |

Table 2 below shows that the most study subjects were housewives and were illiterate 78 (56%). A total of 25 (18%) were middle pass, 17 (12%) had gone to high school, 13 (9%) had gone for higher secondary education, and only 6 (4%) were graduates. Among the controls, most, 29 (29%), held diplomas, 26 (26%) were illiterate, 15 (15%) were graduates, 10 (10%) were middle pass, 8 (8%) had gone to high school, and 9 (9%) had gone for higher secondary education.

**Table 2.** The distributions of patients and controls according to education (*n* signifies the number of subjects and controls).

| Education | Patients (*n* = 140) | Control (*n* = 100) |
|---|---|---|
| Middle | 25 (17.86) | 10 (10.00) |
| High school | 17 (12.14) | 8 (8.00) |
| Hr. secondary | 13 (9.29) | 9 (9.00) |
| Diploma | 0 (0.00) | 29 (29.00) |
| Graduate | 6 (4.29) | 15 (15.00) |
| Post-graduate | 0 (0.00) | 3 (3.00) |
| GNM | 1 (0.71) | 0 (0.00) |
| None | 78 (55.71) | 26 (26.00) |

Among study subjects with the disease, 94 (67%) had a disease duration ≤ 5 years, 26 (19%) had a disease duration between 6 and 10 years, 10 (7%) had a duration of 11–15 years, and 10 (7%) had >10 years disease duration. In the group, 31 (25%) patients had a DAS score ≤ 4.0, 63 (50%) patients had a DAS score between 4.01 and 6.0, and in the remaining 31 (25%) subjects, the DAS score was > 6.0. The mean DAS score among study subjects was 4.96.

The study subjects had a mean ADL score of 11.64, while the controls had a mean score of 2.42 with a statistically significant *p*-value as shown in Table 3.

The global fatigue index (GFI) was higher in the study subjects, with a mean of 33.16, whereas in controls, it was 14.41 with a significant *p*-value.

The global fatigue index showed a significant impact of disease on fatigue parameters when compared with controls as *p*-value showed a very significant difference as shown in Table 4.

**Table 3.** The activities of daily living (ADL) scores in the study subjects and controls.

| Groups | ADL Score Number of Subjects (*n*) = 140, Number of Controls (*c*) = 100 | |
| --- | --- | --- |
| | **Mean ± SD** | ***p*-Value** |
| patients | 11.64 ± 7.45 | <0.0001 |
| Control | 2.42 ± 2.54 | |

**Table 4.** Global fatigue indexes (GFIs) in patients and controls.

| Groups | GFI Number of Subjects (*n*) = 140, Number of Controls (*c*) = 100 | |
| --- | --- | --- |
| | **Mean GFI ± SD** | ***p*-Value** |
| Patients | 33.16 ± 4.24 | <0.0001 |
| Control | 14.41 ±11.36 | |

All four parameters that define the quality of life showed statistically significant results; hence, disease has a clear effect on the quality of life of the study group along with fatigue. Table 5 gives the detailed QOL scores of the two groups.

**Table 5.** Who (QOL) scores of different parameters.

| WHO-QOL Comparison in Subjects and Control Number of Subject (*n*) = 140, Number of Controls (*c*) = 100 | | | |
| --- | --- | --- | --- |
| **Group** | **Patients (Mean ± SD)** | **Control (Mean ± SD)** | ***p* Value** |
| Physical health | 35.44 ± 8.59 | 60.60 ± 16.81 | <0.0001 |
| Psychological health | 39.56 ± 7.74 | 59.86 ± 13.34 | <0.0001 |
| Social relationships | 48.76 ± 6.01 | 59.08 ± 13.47 | <0.0001 |
| Environment health | 42.67 ± 8.65 | 58.66 ± 15.37 | <0.0001 |

The linear correlations between the global fatigue index and quality of life in two groups showed that the global fatigue index had a negative impact on the physical, psychological, and environmental domains of quality of life with a significant *p*-value; it had no impact on social relationships as shown in Table 6 below.

**Table 6.** Correlations of the global fatigue index with the WHO (QOL) domains in patients.

| Variables | Pearson's-r | ***p*-Value** |
| --- | --- | --- |
| Physical health | −0.508 | <0.0001 |
| Psychological health | −0.598 | <0.0001 |
| Social relationships | 0.052 | 0.544 |
| Environment health | −0.314 | <0.0001 |

## 9. Discussion

RA is a potentially devastating condition for patients, especially in the early stages of the disease, producing persistent pain, depression or other psychological distress, impaired physical function, decreased quality of life (QOL), and higher medical and social costs [19]. Severe fatigue is frequent in people with RA, and it has a big influence on their quality of life (QOL). Physicians often overlook RA fatigue, and anti-RA medicines have minimal effect on fatigue [20]. The average ages of the patients and controls in our investigation were nearly identical: The average age of the study subjects was 47 years, whereas the average age of

the controls was roughly 42 years. Only 16% of subjects were male, while 84% were female, resulting in a roughly 1:5 ratio. Although it is generally known that rheumatoid arthritis is mostly a female illness, our findings are consistent with those of Dr. Mir Nadeem et al. [21] in terms of age and gender distribution, with a female preponderance of 83% and the majority of patients in the 30–60 age group. Our findings indicate that RA affects three domains of QOL (physical ($p < 0.0001$), psychological ($p < 0.0001$), and environmental ($p < 0.0001$) but not social ($p = 0.54$) compared with controls, with the physical domain the most affected. Study subjects with RA have significantly poorer QOL than the general population, as reported by Bendtsen et al. [22] and Suurmeijer et al. [23].

The MAF scale was utilized to measure fatigue in our study. The MAF is a RA-specific update of the Piper Fatigue Scale [17]. It was designed in oncology, has demonstrated face and content validity, and was compared with another fatigue scale and found to be a valid and dynamic scale for RA fatigue evaluation. In our study, study subjects had a higher global fatigue index of 33.16, compared with 14.41 in the controls, with a significant *p*-value of 0.0001. The global fatigue index (GFI) exhibited a substantial negative influence on the physical, psychological, and environmental quality of life categories, but it had no effect on social interactions. The mean GFI was substantially higher than that of the general population in research by Schwarzer et al. [24], suggesting that fatigue is a severe issue in RA. Furthermore, our findings matched those of earlier RA patient studies conducted in the Netherlands and Denmark (Loppenthin et al. [25], Rupp et al. [26]). However, social support was high in our study; this discrepancy can be explained by cultural variations, shorter disease duration, fewer abnormalities, and lower social stigma as was seen with parameters of the social domain for WHO-QOL. However, other researchers have indicated that patients with RA always lack appropriate social support (Mcinnes et al.) [27], which is likely due to the fact that RA is incurable and has a protracted course, resulting in deformity and physical handicap in its later stages. Furthermore, most RA patients participate in limited social activities, which contributes to a lack of societal support [28]. As a result, social support may be very important for RA-related fatigue, and it may be able to alleviate it to a great amount. Hence, we recommend that all patients presenting to physicians be evaluated for fatigue; ignoring fatigue in RA patients might result in increased morbidity in the form of poor QOL.

## 10. Limitations

The lack of an internationally established definition of RA fatigue, which is outside the scope of this study, is one of the study's shortcomings. Although our study was one of the first on the Asian subcontinent, it was conducted in north India, and it remains to be seen if the findings can be applied to other people with diverse cultural backgrounds.

The MAF scale is an RA-specific update of the Piper Fatigue Scale that was designed in oncology, has provided face and content validity, and was compared with another fatigue scale and found to be a valid and dynamic scale for RA fatigue. However, the scale might benefit from more study, particularly in terms of content validity for RA patients, with the inclusion of cognitive components and with validating the changes in the sensitivity of the measurement of fatigue in these patients.

## 11. Conclusions

To conclude, in our study, fatigue was a persistent problem, despite treatment. The median level of fatigue experienced by patients with RA was high. Therefore, as persistent fatigue is associated with functional loss, fatigue in RA remains an 'unmet need' and continues to be ignored by clinicians. Our study provides reasonable evidence that fatigue can be used both as a main measure in RA and as a good predictor of physical activity and overall quality of life.

**Author Contributions:** W.R.D.: principal investigator, conceptualization, project administration. I.A.M.: methodology, data validation. M.N.: data curation, writing, investigation, formal analysis. G.S.: supervision, project administration. S.S.: methodology, formal analysis. All authors have read and agreed to the published version of the manuscript.

**Funding:** This research received no external funding.

**Institutional Review Board Statement:** The study was conducted in accordance with the Declaration of Helsinki, and approved by the Institutional Review Board of Government Medical College Jammu under order no. PGRP/2017/MDGMC.

**Informed Consent Statement:** Informed consent was obtained from all subjects involved in the study.

**Data Availability Statement:** Not applicable.

**Conflicts of Interest:** The authors declare no conflict of interest.

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
