# Peer review of "The Assessment of Fatigue in Rheumatoid Arthritis Patients and Its Impact on Their Quality of Life"

_clinpract, doi:10.3390/clinpract12040062_

Round 1
Reviewer 1 Report
orthographic errors, not sufficient definition of control group as well as patient's group (depression etc), what kind of therapy against RA received examined patients, figures 1-2 were not discussed in results etc
Author Response
Respected reviewers, I thank you for considering our article worth for publication and also thanks for such an extensive review of the document which gives us a good idea about writing a good research paper. reply to the author's comments are below
1.orthographic errors--- while addressing the reviewer queries proper care has been taken to address orthographic errors
2. not sufficient definition of the control group as well as the patient's group (depression etc),---- definition of case explained in review manuscript as Patients diagnosed as Rheumatoid Arthritis according to 2010-ACR-EULAR classification criteria for Rheumatoid arthritis aged 16 years and above.
compared to age and sex matched controls attending Medicine/ rheumatology OPD having health conditions other than musculoskeletal complains
3. what kind of therapy against RA received examined patients,--- RA therapy has been mentioned in Patients as they were on treatment for rheumatoid arthritis with DMARDS mainly
figures 1-2 were not discussed in results etc-----figures although discussed in results but as per other reviewers suggestion have been replaced by the single table.
Reviewer 2 Report
The authors analyzed the effect of fatigue on the quality of life in patients with RA.
Comments
1. Lines 17-18: This sentence is not clear (Fatigue predicts… –what?). It should be clarified.
2. Lines 33-34: This sentence is not clear. It should be clarified.
3. All the repeated sentences should be corrected: Lines 61-62 versus 74-75; 64-65 versus 73-74; 108-110 versus 127-128; 180 versus 185 versus 193.
4. GFI abbreviation should be disclosed.
5. Figures and Tables: The number of patients should be indicated in all the illustrations.
6. Fig 2. All the abbreviations should be disclosed and all the columns should be named.
7. Limitations should be placed before conclusions. This should be corrected.
Author Response
thankyou rewiewers for your detailed review of the manuscript and that your good self appreciated our work and considered it suitable for publication . response to the queries are mentioned below ;
- Lines 17-18: This sentence is not clear (Fatigue predicts… –what?). It should be clarified.----dear reviewer it was just a typing error fatigue predicts activity of daily living in RA patients .sentence clarified in text .
- Lines 33-34: This sentence is not clear. It should be clarified.---- clarified the text appropriately
- All the repeated sentences should be corrected: Lines 61-62 versus 74-75; 64-65 versus 73-74; 108-110 versus 127-128; 180 versus 185 versus 193----- corrected as per reviewer advice .
- GFI abbreviation should be disclosed.----- disclosed as per suggestion global fatigue index.
- Figures and Tables: The number of patients should be indicated in all the illustrations.----- number of patients and controls mentioned as per suggestion
- Fig 2. All the abbreviations should be disclosed and all the columns should be named. ---- both first and second figure replaced with tables as per other reviewers suggestion
- Limitations should be placed before conclusions. This should be corrected.---- thanks for suggestion done as advised,
Reviewer 3 Report
Fatigue was previously been well explained in RA. The study significance and scientific strength of this manuscript need critical improvement. The clinical markers for RA were also not been measured by the authors in the characterization of RA in subjects, which is one of the major limitations of the study. Overall, the structure of the manuscript needs major corrections and the results section should be more focused.
- Check the author names correct?
- Line 20 (use past sentence)
- Sentences in the abstract are repeated in the introduction, aim, and result sections which must be corrected.
- The introduction should be more comprehensive and into two paragraphs – explaining RA and fatigue and their correlations in the first paragraph, and MAF and ADL scoring systems and which method is the most applicable, and which method did you adopt with the reason for the same.
- Figs 1 and 2 must be made into a single table with exact numbers and percentages in the brackets.
- Results section was found to be empty and needs the findings in detail.
- Discussion should be also made into separate paragraphs as required.
Author Response
respected reviewer thankyou very much for detailed review it gives us a learning experience in research and paper writing. response to reviewer comments are given below
Fatigue was previously been well explained in RA. The study significance and scientific strength of this manuscript need critical improvement.----- improvement has been done as suggested
The clinical markers for RA were also not been measured by the authors in the characterization of RA in subjects, which is one of the major limitations of the study.----the patient who were already diagnosed with RA have been taken in the study and we have measured the disease activity through DAS 28 score which has been mentioned in manuscript and includes both clinical as well as biochemical markers of disease
Overall, the structure of the manuscript needs major corrections and the results section should be more focused.----proper attention has been given to the suggestion which editing the paper
- Check the author names correct?-----done
- Line 20 (use past sentence)----done as suggested
- Sentences in the abstract are repeated in the introduction, aim, and result sections which must be corrected.------paraphasing done
- The introduction should be more comprehensive and into two paragraphs – explaining RA and fatigue and their correlations in the first paragraph,-------done as suggested
- and MAF and ADL scoring systems and which method is the most applicable, and which method did you adopt with the reason for the same.----MAF multidimensional assessment of fatigue and ADL activity of daily living score are two different scores and both are used in our manuscript and both of the signifies two different parameters . MAF calculates the fatigue dimension and ADL the quality of life in RA patients then both these scores are correlated to see effect of fatigue on quality of life in RA patients
- Figs 1 and 2 must be made into a single table with exact numbers and percentages in the brackets.----done
- Results section was found to be empty and needs the findings in detail.-----uplaoded again after edition
- Discussion should be also made into separate paragraphs as required----- done as per suggestion.
Round 2
Reviewer 3 Report
- Prof in the author's name means "Professor". If so, only the name has to be added.
- For the tables, parameters need to be mentioned clearly (Example: For Table. 1, if it is in percentage or in number?)
- In the whole manuscript, patient must be changed to study subjects or study participants
- Results still need improvements and the Introduction can be more specific
- The manuscript accepting all the track changes needs to be uploaded separately to check the flow of the manuscript. Also, the authors are requested to address the reviewers point by point and mention on which line in the manuscript they have revised the text.
Author Response
once again i on behalf of my coauthors thank the reviewers for their precious time and such detailed review where all the attention was given to even minute issues in our manuscript it gives us learning experience while addressing reviewer comments . hope we will address the comments clearly.
reviewer 3 comments response :
Prof in the author's name means "Professor". If so, only the name has to be added. ------ done as per reviewer suggestion
For the tables, parameters need to be mentioned clearly (Example: For Table. 1, if it is in percentage or in number?)----- parameters are usually represented in numbers with percentage in bracket. same has been mentioned in manuscript
In the whole manuscript, patient must be changed to study subjects or study participants------ done and marked in red
- Results still need improvements and the Introduction can be more specific------ full attention was given to introduction and results formulated in better way and marked in red
- The manuscript accepting all the track changes needs to be uploaded separately to check the flow of the manuscript. Also, the authors are requested to address the reviewers point by point and mention on which line in the manuscript they have revised the text.-----done as per reviewer comments
Round 3
Reviewer 3 Report
The authors need to give proper attention to correcting their manuscript. Many instances were noted with repetition in other sections with the introduction. Still, there is no flow in the presentations in the manuscript. The authors are requested to consult with an academic english language expert before submitting the revised version.
Line 20: is to be changed to was
Line 21: to access incidence to be changed to 'to access the incidence'
Line 28-30: Rewrite the sentence with grammar
Line 81: is to be changed to was
Line 143-144: The words are repeated with the introduction
Line 146-150: The sentences look cloudy and need to be divided into what exactly the authors found. Grammar to be maintained.
Line 152-156: The sentences are repeated with the introduction
Table 3 is not legible. no. and the number is the same.
Line 193: the reference needs more information. Same result???
Author Response
Respected reviewers i am sorry for this mistakes which were found on third review of our manuscript mainly due to grammar and typing errors and as per reviewers suggestion it took us 48hours to get a professional editor to rewrite the paper with proper grammatical considerations. i hope this time we wont disappoint you The authors need to give proper attention to correcting their manuscript. Many instances were noted with repetition in other sections with the introduction. Still, there is no flow in the presentations in the manuscript. The authors are requested to consult with an academic english language expert before submitting the revised version.-----done as per suggestion.
Line 20: is to be changed to was---done
Line 21: to access incidence to be changed to 'to access the incidence'---- done
Line 28-30: Rewrite the sentence with grammar--- done.
Line 81: is to be changed to was----done
Line 143-144: The words are repeated with the introduction--- reviewed the sentences which were repeated hope we were able to accomplish what reviewer wanted
Line 146-150: The sentences look cloudy and need to be divided into what exactly the authors found. Grammar to be maintained.---- done as per reviewer comments
Line 152-156: The sentences are repeated with the introduction----- paraphrased the sentence hope the lines coincide
Table 3 is not legible. no. and the number is the same.--- sorry it was a typing error and corrected same
Line 193: the reference needs more information. Same result???----- the reference shares the same result in term of female preponderance of RA and age and gender distribution